# Zero-Shot Pedestrian Detection in Autonomous Driving Systems under Day vs Night Conditions

Anonymous Full Paper
Submission 42

## Abstract

Autonomous ultralight vehicles operate in varied lighting conditions, where robust pedestrian detection is critical. This paper examines zero-shot pedestrian detection performance across day, twilight, and night scenarios using modern Transformer-based models. We leverage the large-scale BDD100K driving dataset to compare Real-Time DEtection TRansformer (RT-DETR) against its improved successor RT-DETRv2 on identifying pedestrians without any fine-tuning. Our experiments fix IoU at at 0.5 and analyze recall as detection confidence varies. Results show a significant drop in recall from day to night, indicating that low-light conditions degrade detection. RT-DETRv2 consistently outperforms RT-DETR, recovering a portion of missed detections under all lighting conditions. We discuss the implications for deploying these models in ultralight electric utility vehicles (ULEVs) where human operators and vehicles share tasks, highlighting the need for adaptive learning and operator feedback to maintain safety after dark. Future work will integrate interactive learning to improve night-time perception.

## 1 Introduction

Micromobility and ultralight electric utility vehicles (ULEVs) are emerging as practical alternatives to vans and cars for campus services, facility maintenance, short-range logistics, and last-mile delivery in dense urban spaces [1, 2]. Europe's policies are pushing strongly toward zero-emission, human-scaled vehicles and cleaner urban freight, making ULEVs not just desirable but essential [3]. Because ULEVs are compact and slower-moving, they tend to reduce collision risk and lessen the severity of pedestrian injuries compared to heavier or faster vehicles. For instance, an impact at about 27 km/h (17 mph) yields only 10% risk of severe injury for a pedestrian, whereas at 53 km/h (33 mph) the risk jumps to 50% [4]. But this safety benefit depends on pedestrians being reliably detected to begin with. Detection must remain effective in all lighting conditions (day, dawn or dusk, and night) to prevent collisions [5]. Urban streets pose numerous visual challenges, especially under low light: harsh headlight glare, deep shadows, and dim illumination can all undermine computer vision performance and cause pedestrian detection failures after dark. ULEVs' operational mode, where a human operator alternates between driving/riding and walking alongside the vehicle, makes consistent pedestrian detection under varied lighting even more essential [6].

From a data perspective, we have the right ingredients to study this problem without artificial constraints. The BDD100K dataset [7] contains 100,000 driving video frames captured across diverse locations, weather, and lighting conditions, with explicit labels for attributes like time of day. This allows us to directly compare detection performance between day, dusk/dawn, and night scenes using the same dataset. Night-focused collections like NightOwls highlight how much harder detection becomes after dark: they show significantly more missed pedestrians due to issues like motion blur, sensor noise, and uneven illumination from artificial lights and reflections [8]. Likewise, low-light image datasets such as ExDark (Exclusively Dark) cover scenes from extremely dim environments up through twilight, across many object categories, enabling analysis of how illumination alone affects object detection [9]. These resources underscore the challenges we target: detection algorithms struggle as lighting diminishes or becomes inconsistent.

Meanwhile, object detection models have rapidly evolved from convolutional architectures to Transformer-based systems [10]. The DEtection TRansformer (DETR) reframed detection as a direct set prediction problem, using one-to-one matching between predicted objects and ground truth and thus eliminating the need for non-maximum suppression in post-processing [11]. This end-to-end paradigm simplified detection pipelines, but early DETR models had slow convergence and high computational cost, which limited their practicality for real-time use. Real-Time DETR (RT-DETR) addressed this by designing a hybrid Transformer encoder and an uncertainty-minimal query selection mechanism to speed up inference and maintain accuracy [12]. In essence, RT-DETR decouples intra-scale and cross-scale feature processing for efficiency. It selects high-confidence region features as queries for the decoder to improve results. The model achieves $> 100$ FPS on a T4 GPU with a ResNet-50 backbone [13]. It matches or surpasses YOLO-series detectors on the COCO benchmark [12]. The newer RT-DETRv2 builds on its predecessor with a series of accuracy-

enhancing improvements that do not compromise inference speed. It uses different numbers of sampling points at each feature scale in the deformable attention module, improving multi-scale feature representation. To address deployment constraints seen in earlier DETR-based models, it replaces the grid sampling operation with a discrete sampling operator, making the architecture more hardware-friendly. Additionally, it leverages dynamic data augmentation and scale-adaptive hyperparameter tuning during training, boosting detection performance with no cost to runtime [14]. These advances make the DETR family increasingly suitable for embedded and edge platforms like those on ULEVs, which have limited compute but need real-time performance [10, 14].

Using the ULEV use-case as motivation, this paper presents a focused empirical study. We analyze pedestrian detection performance on BDD100K across day, dusk, and night conditions in a zero-shot setting, i.e., employing models pre-trained on general datasets, with no fine-tuning on BDD100K. We compare RT-DETR (ResNet-50vd backbone) against RT-DETRv2 (also ResNet-50vd) using a fixed IoU threshold of 0.5 for evaluation. We vary the detection confidence threshold to examine the precision–recall tradeoff. Our results show that lighting has a major impact on detection recall. For instance, recall drops by roughly 8–13 percentage points from day to night with the same model. Whereas, the improved RT-DETRv2 outperforms RT-DETR consistently in all lighting conditions. We quantify how large the performance gap is in each regime. These findings inform the perception module choices for ULEVs that must operate reliably under different lighting. They also support our long-term plan to integrate imitation learning and interactive human-in-the-loop training by establishing a performance baseline for off-the-shelf detectors. We can later demonstrate how additional online learning with operator feedback can close the gap in night-time performance. This study is part of an ongoing applied collaboration on ULEV perception; further details will be provided in the Acknowledgments.

In summary, our contributions are as follows: (1) We provide a benchmark analysis of pedestrian detection split by time-of-day on the BDD100K dataset, using modern real-time DETR-based models, to highlight how changes in illumination affect detection in a realistic autonomous driving setting relevant to ULEVs. (2) We discuss deployment implications for collaborative autonomy in ULEVs, where a human operator is in the loop, pointing out that perception must remain robust in low-light for safety, and suggesting how an interactive learning approach could help. (3) We position these results as a baseline for the perception module in a larger human–machine collaboration project, to be extended with conditional imitation learning and interactive feedback as outlined in our project roadmap.

We proceed as follows. Section 2 describes the dataset and how we partitioned it by time of day. Section 3 outlines our methodology including the models and evaluation protocol. Section 4 presents the results and critical analysis, and Section 5 discusses implications and limitations. Finally, Section 6 concludes the paper and sketches future directions, including integrating learning-on-the-fly to improve night-time detection.

## 2 Dataset

We base our study on the BDD100K dataset, a large and diverse driving dataset released by Berkeley DeepDrive [7]. BDD100K comprises 100,000 video clips recorded from driving in various U.S. locations, spanning many kinds of weather, lighting, and scene types. The dataset features frames extracted from those videos, annotated for tasks like object detection, segmentation, and lane markings. It supports ten different tasks from object detection to lane following and segmentation and provides rich annotations. Crucially for our purposes, each image in BDD100K has associated frame-level attributes. We utilize the *timeofday* attribute, which labels each frame as daytime, dawn/dusk, or night, allowing us to create subsets of the data according to time of day for controlled comparisons.

For the object detection task, BDD100K provides "Detection 2020" annotations in a JSON format with each object annotated by a bounding box and class label, and image-level attributes included. We use the official detection annotations for both training and validation sets. Specifically, we parsed the JSON files and filtered images by the *timeofday* field. This yielded three splits of interest: one set of daytime images, another of dawn/dusk images, and a night set. To enable rapid experimentation such as finding confidence thresholds and comparing models without excessive computation, we constructed capped subsets of each split. We randomly sampled up to 1500 images from each time-of-day category in the BDD100K training set. In doing so, we skipped 139 training images whose time-of-day attribute was missing or "undefined," and we also skipped any image for which the corresponding image file was not found. This sampling strategy gave us three like-for-like subsets, each $\leq$ 1500 images, for initial analysis and parameter tuning, while keeping the dataset's ontology, camera viewpoint, and annotation format consistent across conditions.

We report most of our ablation and sensitivity results on those equalized train subsets. After determining an appropriate confidence threshold using the train subsets, we then evaluate the selected models on the full validation set for each time of day,

using all available validation images. Finally we report the main results on those full validation sets. For the validation evaluation, we do not cap the number of images. We include all images labeled as day, dusk, or night in BDD100K validation set. We also report coverage, which is the count of images that have at least one pedestrian versus the total number of images in each partition. This is important because the share of "empty" frames with no pedestrians changes with time of day. This affects the interpretation of recall. For example, night scenes in BDD100K tend to have fewer pedestrians per image on average, and also a larger share of images with none at all (many highway or low-activity night shots). All evaluation is done on the validation set because BDD100K's test set labels are withheld for a challenge server. Thus, our threshold experimentation and recall numbers are reported on val, which is the appropriate offline validation benchmark.

## 3 Methodology

Our goal is to evaluate pedestrian detection in a zero-shot scenario across different lighting conditions. "Zero-shot" here means we take pre-trained detection models as-is, without any fine-tuning on BDD100K or any specific night-time data, and directly test their performance on the dataset splits. This isolates the effect of lighting on the model's inherent generalization ability. We focus on two recent transformer-based detectors: RT-DETR (Real-Time Detection Transformer) and RT-DETRv2, both using a ResNet-50-vd backbone. These models were chosen for their relevance to real-time operation on edge devices. RT-DETR and RT-DETRv2 are the state-of-the-art real-time transformers introduced by Zhao et al. [12] and Lv et al. [14] respectively, with RT-DETRv2 building on RT-DETR's design as described earlier. We obtained the models from public checkpoints (pre-trained on COCO [15]). No additional training or domain adaptation was performed on BDD100K. This is a strict test of how well a generic detector can perform on unseen data from a different distribution.

***Evaluation protocol:*** We treat it as a standard object detection evaluation focusing on the "person" class (pedestrians). Detections are counted as True Positives (TPs) if they overlap a ground-truth pedestrian with IoU $\leq 0.5$ and if the detection is assigned the correct class (person). Duplicates or false positives are ignored for recall calculation. Each ground truth can match only one true positive. We measure Recall = TP / (TP + FN), i.e. the fraction of ground-truth pedestrians that were detected. Because missing a pedestrian can lead to serious safety failures we make recall our primary metric. We also report FNR (false negative rate = 1 – Recall) interchangeably. We do not place emphasis on precision in this zero-shot analysis, because the detectors were not fine-tuned to balance precision on this dataset and because in a safety context we would likely tolerate some false alarms as long as we minimize missed detections. However, we do study how varying the confidence score threshold for detections impacts recall, which indirectly reflects the precision trade-off. Specifically, we apply three confidence thresholds (0.3, 0.5, 0.7) to the model outputs to see how many detections are retained and how many true positives are hit or missed at each setting.

We first ran both RT-DETR and RT-DETRv2 on the capped train subsets (1500 images each category) to get initial performance numbers and to choose a reasonable confidence threshold. The models output bounding boxes with confidence scores; by default we considered a threshold of 0.5 (common in literature for reporting "Recall@IoU=0.5" at a fixed score cut-off [15–17]). We then tried a lower confidence threshold of 0.3 to see whether recall would increase, even though that might bring in more false positives. We also tried a higher threshold of 0.7 to test how many true positives would be lost if the system were more conservative. These results guided us in setting the threshold for the final evaluation. After confirming the threshold choice, we evaluated both models on the full validation set splits; all day, all dusk, all night images in validation set. We report the recall and FNR for each model under each lighting condition. Additionally, we note the number of pedestrian instances and images in each split to provide context.

All inference are performed on a single Apple M4 Max workstation with 36 GB unified memory. The RT-DETR model produces a fixed set of predictions per image, equal to the number of object queries in its decoder. Non-maximum suppression is not used. The models internally handle duplicate removal via their set matching loss. So the score threshold mainly serves to filter out low-confidence predictions which are likely noise. Importantly, because we are comparing two models, we hold the evaluation criteria constant for both, same IoU threshold and same confidence threshold, to ensure fairness.

## 4 Results & Critical Analysis

***Baseline performance on train subsets (RT-DETR):*** We first examine how the original RT-DETR model performs under different lighting using the sampled train splits and a 0.5 confidence threshold. Table 1 summarizes the recall for the person class in each subset. Out of 1500 daytime images, 615 contained at least one pedestrian (total 2718 ground-truth persons); RT-DETR detected 1205 of these, yielding a recall of 0.443 (44.3%) and

| Split | Images/1500 | GT | TP | FN | Recall | FNR |
|-------|-------------|------|------|------|--------|-------|
| Day | 615 | 2718 | 1205 | 1513 | **0.443** | 0.557 |
| Dawn/Dusk | 498 | 2018 | 852 | 1166 | **0.422** | 0.578 |
| Night | 309 | 1034 | 370 | 664 | **0.358** | 0.642 |

**Table 1.** RT-DETR-R50 (COCO-pretrained), BDD100K train split, Recall@IoU=0.5 (person).

FNR about 0.557. For dawn/dusk images, recall was slightly lower at 0.422. The night subset was the most challenging: only 309 of 1500 night images had persons (1034 total people), and RT-DETR caught 370 of them. Recall 0.358 (35.8%) means nearly 64% of pedestrians at night went undetected by this model. This preliminary result confirms a clear drop in detection ability at night. The gap from 0.443 (day) to 0.358 (night) is substantial in safety terms. The model is missing almost two-thirds of pedestrians in dark conditions. Dusk/dawn sits in between (42.2% recall), as expected, twilight lighting is not as difficult as full darkness, but still worse than broad daylight.

***Effect of score threshold:*** One might wonder if the model actually "sees" more pedestrians at night but gives them low confidence scores, which could be fixed by lowering the threshold. To test this, we evaluated recall at thresholds 0.3 and 0.7 on the same data (Table 2). Interestingly, lowering the confidence cut from 0.5 to 0.3 did not yield any new true positives for any time-of-day split. The recall remained exactly the same (e.g. 0.358 at night). This suggests that most of the correct detections were already scored above 0.5. Any additional predictions between 0.3 and 0.5 confidence were either duplicates or false alarms that did not correspond to real pedestrians. In other words, RT-DETR's low-confidence predictions didn't help recover missed people. On the other hand, raising the threshold to 0.7 had a strong negative impact, especially at night. Daytime recall fell to 0.292 (a relative drop of 34%), and night recall fell to 0.182 (only 18.2% of night-time pedestrians detected at high confidence), see Figure 1. This indicates that at night many of the detections that are correct are relatively low confidence (in the 0.5 range); making the detector overly strict wipes out half of the true positives. For our purposes, missing fewer pedestrians is paramount, so a threshold significantly above 0.5 would be inadvisable. We therefore chose to stick with 0.5 as the operating point for subsequent comparisons, since lowering didn't help and higher would sacrifice too much recall.

***RT-DETR vs. RT-DETRv2 (train subset comparison):*** Table 3 presents a side-by-side comparison of the two models on the 1500-image train splits at the chosen 0.5 threshold. We see that RT-DETRv2 consistently outperforms the original RT-DETR in recall for all lighting conditions. In

| Split | score_thr | Recall | FNR |
|-------|-----------|--------|-------|
| Day | 0.3 | 0.443 | 0.557 |
| Day | 0.5 | 0.443 | 0.557 |
| Day | 0.7 | 0.292 | 0.708 |
| Dawn/Dusk | 0.3 | 0.422 | 0.578 |
| Dawn/Dusk | 0.5 | 0.422 | 0.578 |
| Dawn/Dusk | 0.7 | 0.266 | 0.734 |
| Night | 0.3 | 0.358 | 0.642 |
| Night | 0.5 | 0.358 | 0.642 |
| Night | 0.7 | 0.182 | 0.818 |

**Table 2.** Recall@IoU=0.5 by confidence threshold. Lowering from 0.5 to 0.3 did not add new matched TPs; raising to 0.7 hurts recall, especially at night.

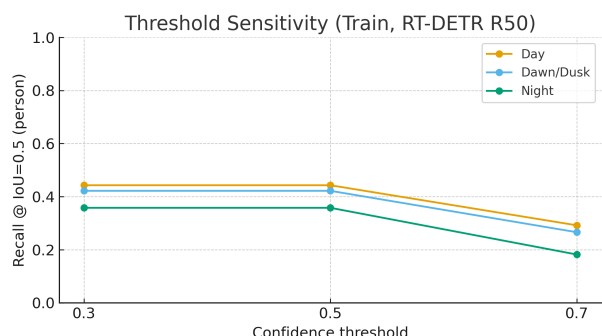

**Figure 1.** Threshold sensitivity on BDD100K train (1,500 imgs/split), RT-DETR-R50 (COCO-pretrained), person class: Recall@IoU=0.5 for day, dawn/dusk, night as confidence threshold changes. Lowering from 0.5 to 0.3 adds nothing, raising to 0.7 drops recall, especially at night.

daytime, RT-DETRv2 achieves 0.504 recall (about 50.4%), which is an absolute improvement of ∼6.1 percentage points over RT-DETR's 0.443. In dusk/dawn, v2 reaches 0.479 recall vs. v1's 0.422 (a +5.7 point gain). At night, v2 manages 0.442 recall, notably higher than v1's 0.358 (+8.4 points, which is over 23% relative improvement). In terms of missed detection rate (FNR), RT-DETRv2 brings the nighttime miss rate down from 64.2% to 55.8%. These improvements confirm that the enhancements in RT-DETRv2 (better multi-scale feature sampling, training augmentation, etc.) are yielding tangible benefits for pedestrian detection, especially under poor lighting. However, even with RT-DETRv2, the recall at night is still only ∼44%, meaning more than half of pedestrians are missed in dark scenes. So

| Model | Day | | Dawn/Dusk | | Night | |
|---|---|---|---|---|---|---|
| | Recall | FNR | Recall | FNR | Recall | FNR |
| RT-DETR (R50) | 0.443 | 0.557 | 0.422 | 0.578 | 0.358 | 0.642 |
| RT-DETR**v2** (R50) | **0.504** | **0.496** | **0.479** | **0.521** | **0.442** | **0.558** |

**Table 3.** Recall@IoU=0.5 (person) on BDD100K (train) split by time-of-day.

while v2 is better, the lighting gap remains. Daylight recall (50.4%) is higher than night (44.2%) by about 6 percentage points on v2 (for v1 the gap was ∼8.5 points). This hints that additional strategies such as fine-tuning, data augmentation, or specialized sensors would be needed to further close the night vs day performance difference.

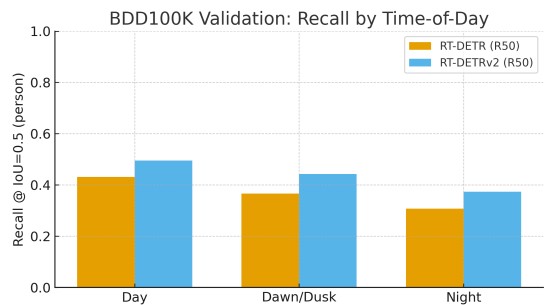

**Figure 2.** Comparison of recall by time-of-day for RT-DETR and its successor RT-DETRv2 on the BDD100K validation set.

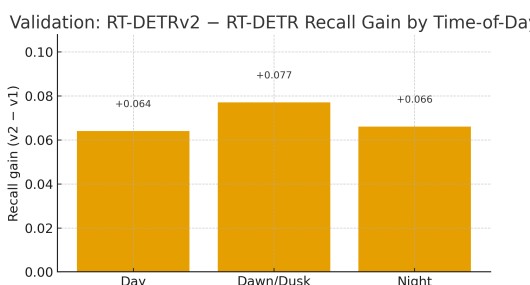

**Figure 3.** Improvement in Recall by time of day (v2 over v1) for RT-DETR models.

***Validation set results:*** Finally, we evaluated both models on the full BDD100K validation images for each time of day, using the same threshold 0.5. Table 4 summarizes these results. We note first that the absolute recall values on validation set are slightly lower than on the train subset, which is expected because the train subset recall was not on training data. We used train images but the models were not trained on them, so it was essentially another test. But the validation set may have different scene distributions or more difficult instances. For RT-DETR, we see Recall = 0.431 day, 0.366 dusk, 0.308 night on the validation set. The trend holds: ∼43% of pedestrians detected in daytime, dropping

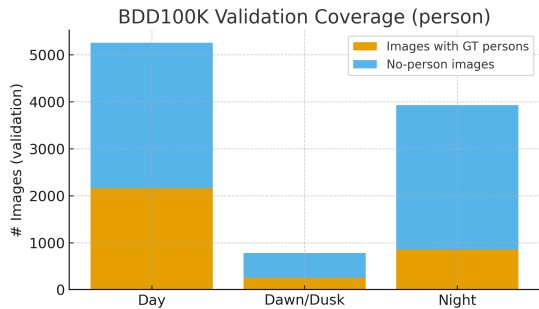

**Figure 4.** BDD100K validation coverage for the person class. Stacked bars show images with at least one GT person (orange) vs. no-person images (blue) per time-of-day: Day 2155/5258, Dawn/Dusk 250/778, Night 853/3929. Coverage differs markedly across splits; recall/FNR are computed only over images containing GT persons.

to ∼30.8% at night. For RT-DETRv2 on validation set: 0.495 day, 0.443 dusk, 0.374 night. The improved model again shows higher recall in each condition (gains of ∼6–7 points), and its night recall is 0.374 (versus 0.308 for v1), , see Figure 3 and 2 for reference. The gap between day and night in absolute terms is ∼12 percentage points for both models on validation set. This is quite consistent with what we observed in the train-like sample. It reinforces that darkness significantly degrades the effectiveness of these vision models in a zero-shot scenario. We also note that dusk/dawn ("twilight") performance is intermediate: v1 had 0.366 recall at dusk, and v2 had 0.443, almost equal to v2's daytime performance of 0.495. This suggests dusk scenes in BDD100K are somewhat closer to daytime in detectability, perhaps due to street lighting or remaining ambient light, though still a bit lower.

To contextualize these numbers, Table 5 provides the dataset statistics for the validation splits. Daytime in validation set had 5258 images, of which 2155 ( 41%) contained at least one person, with a total of 9476 labeled pedestrians. Nighttime had 3929 images, but only 853 ( 22%) had any persons (2882 total pedestrians). Dawn/dusk had the fewest images (778) and 250 with persons (1060 total) (see Figure 4). This shows that the density of pedestrians in the data differs: daytime images often have multiple pedestrians (on average  4.4 per image that has any), while night images have fewer when they do have some (≈ 3.4 per image with pedestrians). Also,

| Model | Day Recall | Day FNR | Dawn/Dusk Recall | Dawn/Dusk FNR | Night Recall | Night FNR |
|---|---|---|---|---|---|---|
| RT-DETR (R50) | 0.431 | 0.569 | 0.366 | 0.634 | 0.308 | 0.692 |
| RT-DETRv2 (R50) | **0.495** | **0.505** | **0.443** | **0.557** | **0.374** | **0.626** |

**Table 4.** BDD100K *val*, Recall@IoU=0.5 (person) by time-of-day.

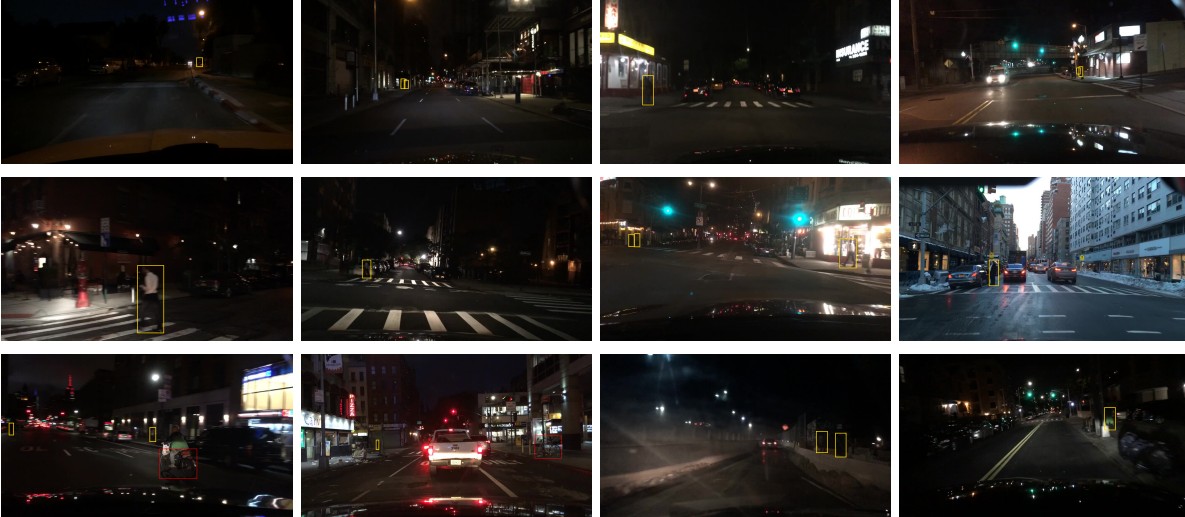

**Figure 5. Night-time failure cases on BDD100K (*val*), person class.** At the operating point (IoU=0.5, score=0.5), RT-DETRv2-R50 produced no correct pedestrian detections in these frames; each panel contains ground-truth pedestrians that were missed and false positives only. Typical causes include low luminance and distance, headlight/backlight glare and reflections, and motion blur.

| Split | Images GT / Total | GT persons |
|---|---|---|
| Day | 2155 / 5258 | 9476 |
| Dawn/Dusk | 250 / 778 | 1060 |
| Night | 853 / 3929 | 2882 |

**Table 5.** BDD100K *val* coverage for the **person** class.

a majority of night frames are completely empty of pedestrians (which could be highway driving at night, etc.). This difference in scene content means that a system operating at night will not only face lower detector recall, but also fewer opportunities (fewer targets), which might be good (less crowding to worry about) or bad (long stretches where the system sees no pedestrians and might become less attentive). In any case, our recall metric is computed only over frames that do contain pedestrians, so it is a fair comparison of detection difficulty.

***Qualitative failure analysis:*** In the most challenging night-time scenarios, our zero-shot pedestrian detector still misses pedestrians despite RT-DETRv2-R50's overall improved performance. Figure 5 gathers 12 representative night-driving images where the model yields no correct pedestrian detections at a confidence threshold of 0.5. Instead, each scene contains only unmatched ground-truth pedestrian boxes (yellow, indicating missed detections)

and false positive boxes (red, marking detections of non-pedestrians). Visual inspection reveals recurring failure modes. Pedestrians often blend into dark backgrounds or are concealed in poor illumination, providing little contrast for the detector. In some situations, oncoming headlights overwhelm the scene. This causes overexposure and lens flare artifacts, which wash out or distort the pedestrian's silhouette. Fast motion further contributes by causing motion blur and obscuring critical features. Several missed pedestrians are also very small or distant. Meaning they occupy only a few pixels and lack clear shape cues. These factors frequently push the model's confidence below 0.5 even when a person is present, resulting in a missed detection. Despite its advances, RT-DETRv2-R50 still struggles in extreme night lighting and imaging conditions. Pedestrians can become nearly invisible to the model or be confused with background clutter. This qualitative failure analysis highlights why edge-case scenarios remain difficult. A combination of low visibility, glare, and motion blur severely reduces the visual cues required for reliable detection, resulting in false negatives and false positives even in today's state-of-the-art systems.

In summary, our results quantitatively confirm that: (a) Lighting conditions have a large impact on off-the-shelf pedestrian detectors – recall drops by

about 25%–30% in absolute terms from daytime to nighttime in our tests. (b) RT-DETRv2 is superior to RT-DETR in this zero-shot evaluation, managing to detect roughly 10% more of the pedestrians at night (and similarly more by day). (c) Threshold tuning is non-trivial making the detector more sensitive did not recover missing detections, which implies the model truly did not produce a detectable output for those pedestrians, rather than just scoring them low. This points to intrinsic limitations in the model's feature representations under night conditions, not merely a confidence issue.

# 5    Discussion

The above findings carry several implications for the deployment of pedestrian detection in autonomous ULEVs and similar platforms. First and foremost, the significant performance degradation at night is a concern for safety. In our zero-shot tests, even the better model (RT-DETRv2) misses about 60% of pedestrians in night images (FNR ∼0.626 on validation set). For a vehicle that is supposed to safely operate in mixed traffic or campus environments, this level of missed detections is unacceptable if the vision system is the primary means of sensing pedestrians. In a practical ULEV setting, one would certainly need to improve this via some combination of (a) model fine-tuning on night-time data, (b) adding other sensors (thermal cameras or LiDAR can help detect pedestrians in darkness), or (c) using active illumination and reflectors. Our current study established the baseline without such enhancements. The next step will be to investigate how much improvement we can get with additional training. The fact that RT-DETRv2 outperforms RT-DETR hints that optimizing the model architecture and training (even still on generic datasets like COCO) yields gains. So training on a more targeted dataset like including NightOwls or dark scenes from BDD100K might substantially raise recall at night.

Another implication is the value of human oversight and interactive learning in the loop. Since we envision ULEVs are semi-autonomous with a human operator nearby, we can leverage that. For example, if the operator is walking and sees a pedestrian that the vehicle did not detect, they could give a corrective cue (verbally or via a gesture interface). Over time, an interactive learning system could accumulate such feedback and adjust the detector. This is part of our project's broader aim to integrate conditional commands and imitation learning. The zero-shot result tells us what the starting point is: even a solid model like RT-DETRv2 will need help to reach the required reliability in dark conditions. The operator could also mitigate risk by taking more manual control in difficult conditions (for instance, driving slower or in tele-operation mode at night or when vision is impaired), but the ultimate goal is to improve the automation to a level where it can be trusted more widely.

Precision vs. recall trade-off is also worth discussing. We focused on recall (sensitivity) because missing a pedestrian is the worst error. The downside of pushing for high recall is false positives, e.g. the detector might incorrectly classify a shadow or a sign as a person. In an interactive human-in-the-loop system, false positives are less dangerous. They might cause a vehicle to slow or stop unnecessarily, which is annoying but not catastrophic and can be corrected by the human. However, too many false alarms could erode the human's trust or attention. Our threshold experiment indicated RT-DETR models at 0.5 are already reasonably balanced. Lowering to 0.3 didn't help recall, and would surely increase false alarms. We might consider adaptive thresholds For instance, perhaps at night accept slightly lower confidence if any detection occurs, since we know recall is generally lower at night. Or use context, if the vehicle is stationary or moving slowly, maybe be more liberal in detecting. These nuanced strategies are beyond the scope of this paper but represent possible deployment heuristics.

It's important to note some limitations of our study. We used only one dataset (BDD100K) for analysis, and while it is large and diverse, it might not capture all nuances of ULEV operation environments, e.g., a campus or industrial site might have different lighting than typical urban streets. Also, our evaluation was purely on vision. As mentioned, many autonomous systems would fuse vision with other sensors. A multi-sensor system could achieve higher detection rates. For instance, thermal imaging can spot pedestrians by their heat signature even in darkness. We also did not explore image enhancement techniques, there is a body of work on improving low-light images via brightening algorithms or noise reduction which could be applied as a pre-processing step to help the detector at night [18]. Another limitation is that we only considered one class, pedestrian. ULEVs might also need to detect cyclists, pets, or other obstacles. It's plausible that night conditions similarly affect those classes, but pedestrians are arguably most critical.

Finally, our recall metric at IoU 0.5 does not account for localization precision. A detection could overlap a pedestrian but still be somewhat off. We assumed that a 0.5 IoU is sufficient for a hit in terms of alerting the vehicle to a hazard. In practical terms, a slightly off-center bounding box is not a big issue as long as the system knows there is something to avoid in that area. For ULEVs moving at low speeds, timely detection is more important than perfect localization.

In conclusion, the results underline a clear challenge, vision models lose a lot of sensitivity in dark

conditions, and while newer architectures improve this to a degree, more work is needed to ensure safe operation. In the next section, we outline how we intend to tackle this through further research.

# 6  Conclusion and Future Directions

We presented an empirical study of zero-shot pedestrian detection performance under varying lighting conditions (day vs. twilight vs. night) using two cutting-edge real-time Transformer models. Our experiments on the BDD100K dataset showed that detection recall for pedestrians drops markedly in low-light images, confirming the intuition that darkness and difficult illumination pose a significant hurdle for vision-based autonomy. RT-DETRv2 demonstrated improved robustness over the original RT-DETR, achieving higher recall across all lighting conditions—about 50% recall by day and 37% by night on the BDD100K validation set, compared to 43% and 31% for RT-DETR. However, even the improved model misses a large fraction of pedestrians at night, highlighting an urgent gap if such models were to be deployed directly in autonomous driving systems.

The analysis in this paper serves as a baseline measurement to guide further developments. As part of our collaborative ULEV project, these findings inform several next steps. One immediate future work direction is to incorporate domain-specific fine-tuning. For example, training the detector on a mix of BDD100K and NightOwls data or applying synthetic brightness augmentation to improve night detection. We expect that fine-tuning would significantly raise the night-time recall (at some cost to precision that we will monitor). Another direction is exploring multi-modal sensing, combining the RGB camera detector with a thermal camera or depth sensor to catch pedestrians that the regular camera misses. Furthermore, we plan to implement an interactive learning loop where the human operator can correct or confirm detections (via voice commands or a tablet interface) and those corrections continuously update the model or its threshold policy. This could take the form of an on-device active learning, where false negatives identified by the human are quickly turned into new training examples perhaps leveraging few-shot learning techniques to gradually improve the model's performance in real time.

In summary, robust pedestrian detection at night remains a challenging problem, but our work quantifies how far current real-time detectors have come and how far they still have to go. By combining improved models like RT-DETRv2 with fine-tuning on relevant data and human-in-the-loop adaptation, we aim to bridge the day–night performance gap. Ensuring that ULEVs can see pedestrians reliably in all conditions is a critical step toward safe autonomous operation in urban and campus environments. We hope this study and the proposed future enhancements will contribute to safer micromobility and zero-emission transport systems, where humans and machines work together seamlessly regardless of the time of day.

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
