# OpenReview forum: "Zero-Shot Pedestrian Detection in Autonomous Driving Systems under Day vs Night Conditions"
_NLDL.org/2026/Conference — Submitted to NLDL 2026_

### Official Review · Reviewer_VMRN · 2025-09-25
**Clarifying the novelty of the contributions is necessary.**

**Rating:** 2
**Confidence:** 4
**Final Rating:** 2
**Final Confidence:** 5

**Summary:**

This manuscript evaluates the pedestrian detection performance of two off-the-shelf detection models, RT-DETR and RT-DETRv2, with a particular focus on varying illumination conditions, such as daytime and nighttime.
The experiments reveal that both models exhibit noticeably lower recall in low-light (nighttime) scenarios compared to daytime.

**Strengths:**

- Investigating pedestrian detection under different lighting conditions is highly relevant for developing reliable autonomous driving systems.

**Weaknesses:**

- W1: The novelty of the findings is unclear, primarily due to the absence of a related work section. The introduction acknowledges that some previous studies have pointed out the lower pedestrian detection performance under low-light conditions, which makes the presented results appear trivial. Related work section should be included to discuss what previous works have unveiled about pedestrian detection, what remains unknown, and how this study contributes. For instance, several prior works, such as [a] and [b], have already explored low-light pedestrian detection. A more extensive discussion would better highlight the uniqueness, positioning, and value of this manuscript.
- W2: The motivation behind the experimental design is unclear. Specifically, it is not justified why both the training set and validation set are used for evaluation.
- W3: Figures are redundant; Figs. 1-4 are simply visual duplicates of Tabs. 1-5 and do not add any new insights.
- W4: For evaluating discriminability of the models between positive and negative samples, ROC-AUC can be another useful metric as it does not depend on the confidence threshold.

[a] Ghari et al., Pedestrian detection in low-light conditions: A comprehensive survey, Image and Vision Computing, Volume 148, 2024.
[b] Yao, He, et al. Nighttime pedestrian detection based on Fore-Background contrast learning, Knowledge-Based Systems 275 (2023).

**Final Justification:**

Considering the other reviews, I would like to keep my score since there is no rebuttal.

**Justification:**

As highlighted in weakness W1, the manuscript lacks clarity regarding its novel contributions. In its current form, it reads more like a technical report than a research paper.

---

### Official Review · Reviewer_gWgz · 2025-10-06
**Review Paper 62**

**Rating:** 1
**Confidence:** 4
**Final Rating:** 1
**Final Confidence:** 4

**Summary:**

Autonomous ultralight electric utility vehicles (ULEVs) must operate reliably under diverse lighting conditions, making robust pedestrian detection essential. This paper evaluates zero-shot pedestrian detection across day, twilight, and night scenarios using Transformer-based detectors. Leveraging the large-scale BDD100K dataset, they compare Real-Time DEtection TRansformer (RT-DETR) with its successor RT-DETRv2, assessing recall at a fixed IoU of 0.5 as detection confidence varies. The results reveal a sharp decline in recall from day to night, underscoring the challenge of low-light conditions. RT-DETRv2 consistently outperforms RT-DETR, recovering a portion of missed detections across all scenarios. The authors discuss the implications for deploying such models in ULEVs, where safe human–vehicle collaboration depends on reliable night-time perception. Finally, they outline future directions toward adaptive and interactive learning strategies to mitigate performance drops in dark.

**Strengths:**

- The authors address an important topic, and I understand why people are important in ULEVs (and not all other classes), as they are often in campus environments.
- The introduction provides a good background to the topic and explains its importance. The reasons for using DETR models are well stated.
- The paper is very easy to read and follow.
- The next steps outlined by the authors appear appropriate and reasonable.

**Weaknesses:**

- The authors claim to be conducting a benchmark analysis, but I find one dataset and only two networks of a similar model to be insufficient for this purpose.
- It is only evaluated on the BDD100K dataset, although two other suitable datasets are mentioned in the introduction.
- Why weren't other backbones such as CSP-DarkNet or ConvNeXt also tested?
- Why split the dataset into training and validation, when the models are applied zero shot and no parameters are actually determined or retrained - so far, there are only results depending on confidence thresholds.

- It is very difficult that precision was ignored in the evaluation. You can increase the number of predictions so much that you no longer overlook any pedestrians, but then you have far too many false positives - which also makes no sense if the vehicle only stops. So there are metrics such as F1 score, which is often used and correlates precision and recall with each other.
- Why are only three confidence thresholds applied? More thresholds could be used, and then metrics such as AUPRC could be determined.
- For a fair comparison, identical IoU thresholds and confidence thresholds are used. I completely agree with the IoU, but for the confidence thresholds, it would be better to use AUC curves and compare the performance based on those.
- "Any additional predictions between 0.3 and 0.5 confidence were either duplicates or false alarms" You don't even find out if there are additional FPs, as these are not examined.
- "RT-DETR’s low confidence predictions didn’t help recover missed people" It is not possible to know, as only one threshold value (0.3) below 0.5 was tested.
- Why a threshold of 0.3?  If I see that the threshold is still too high at 0.3, i.e., no change, then I will test an even lower value.
- "raising the threshold to 0.7 had a strong negative impact" It's obvious that this will happen, as increasing the score only helps to reduce FPs but does not help to detect more pedestrians.
- "We therefore chose to stick with 0.5 as the operating point for subsequent comparisons" Only three different thresholds are tested on one dataset and one network, and then a decision is made to conduct further investigations with a threshold of 0.5. This could be different for the other model.
- "Threshold tuning is non-trivial making the detector more sensitive did not recover missing detections, which implies the model truly did not produce a detectable output for those pedestrians, rather than just scoring them low." This conclusion is not clear from the results.

- I am missing a related work section that deals with pedestrian detection. Are there methods that work well, especially at different times of day?

- The authors use training splits with images less than or equal to 1500. I would rather say how many are available at a minimum or a range.
- Section 2 only states that all available validation images are used, but the numbers are only mentioned in the results. I would add this to section 2.
- "ULEVs might also need to detect cyclists, pets, or other obstacles. It’s plausible that night conditions similarly affect those classes, but pedestrians are arguably most critical." Cyclists are people too, so what's the difference between them and pedestrians?
- Figures 2 and 3 are redundant and show the same information, so it would be better to use the space for other insights.
- Some typos, e.g. line 011 "IoU at at", line 413 "for v1), ,"
- "As part of our collaborative ULEV project" Although the submission is anonymous, I would phrase such sentences in a more general way, as otherwise there could be references to the project.

**Final Justification:**

There was no author rebuttal, and based on the other reviews, I'm sticking with my score.

**Justification:**

The paper is a good basis for future work, but there is not enough content for a benchmark analysis. The analysis has several gaps, not only in the limited datasets/models, but especially in the evaluation, which I consider incomplete. Some points are not well justified, such as splitting the dataset into train and val, as well as the few tested confidence thresholds, and some conclusions.

---

### Official Review · Reviewer_xSMA · 2025-10-07
**Good evaluation of a specific problem for autonomous driving**

**Rating:** 4
**Confidence:** 4
**Final Rating:** 2
**Final Confidence:** 3

**Summary:**

This paper contains an analysis of object detector performance in the context of autonomous vehicles for pedestrian detection in night time scenarios. Since this is an important use-case that needs to be solved, it contains a useful evaluation that shows the gains that have been made regarding pedestrian detection at night, but it also shows that the problem is not yet fully solved.

The paper contains a comparison of RT-DETR and RT-DETRv2 models for pedestrian detection on the BDD100K dataset. It aims to investigate the influence of the time-of-day of the image (daytime / dawn / dusk / night) on the detection performance.

**Strengths:**

Good written paper that analyses thoroughly the detection performance of RT-DETR and RT-DETRv2 of pedestrians in difficult detection (time of day) scenarios. Detection of pedestrians at night remains a challenging problem. It is certainly an area where object detectors still need to improve such that autonomous driving systems or other ADAS systems can assist the driver more reliably.

Considering the above, this paper could make an interesting addition in the literature because it contains a proper analysis of the current SOTA and identifies existing problems that needs to be tackled in the future by the research community.

**Weaknesses:**

The motivation to first evaluate on capped subsets could be elaborated more. I don't yet understand fully why you first need to determine a good confidence threshold on a subset before performing evaluation on the entire dataset.
The entire analysis could be interesting for different confidence thresholds. Is the reason computation time?
Considering the entire dataset immediately will also avoid the discussion on why the final validation has significant different results (e.g. for the night scenes of RT-DETRv2  we have a recall of 0.374 for the entire dataset vs 0.442 for the subset). Does the mean the random sampling was not really random? Is the sampling size of 1500 too small?

Furthermore, it could also be of interest to include more thresholds. E.g., in the examples shown (Fig.5) is there still no evidence of a pedestrian at confidence 0.1 or 0.2?


Some other small remarks:
Line 257: you mean IoU >= 0.5?
Line 422: there is still a difference of 0.05 between recall performance. I would not label that as almost equal.
Figure 5: the image in the right-column looks like it is taken at daytime. Is this image wrongly labelled in the dataset?
Line 575: I would consider mentioning sensor fusion also in the introduction.

**Final Justification:**

Considering there is no rebuttal I would not incline to accept it anymore since my concerns were not addressed.
I agree also with the other reviewers that there are serious questions that need to be answered before this paper can be accepted.

**Justification:**

Overall the paper makes a valuable contribution and i would incline to accept it.

---

### Official Review · Reviewer_6S3e · 2025-10-07
**Zero-Shot Pedestrian Detection with RT-DETRv2 on BDD100K: Confirming Low-Light Performance Trends**

**Rating:** 2
**Confidence:** 4

**Summary:**

This paper evaluates zero-shot pedestrian detection across day, dusk, and night scenarios using two transformer-based object detectors, RT-DETR and RT-DETRv2, on the BDD100K dataset. The authors focus on comparing recall under different detection score thresholds (0.3,0.5,0.7) with a fixed IoU of 0.5. Their results show that both models’ recall drops substantially in night scenes, while RT-DETRv2 consistently outperforms its predecessor. The paper motivates the study by linking performance degradation in low-light conditions to safety concerns for ultralight electric vehicles (ULEVs) operating autonomously or semi-autonomously.

While the study highlights a relevant challenge — robustness of pedestrian detection under varying illumination — it mainly reproduces expected patterns (daytime > dusk > night) without adding novel methodology, analytical insight, or systematic error characterization. The results are descriptive rather than explanatory. The lack of details on preprocessing, threshold selection rationale, and model configurations makes it difficult to reproduce or verify the findings.

**Strengths:**

Timely topic: Pedestrian detection robustness in low-light scenarios is an important and active research area, particularly for safety-critical domains like micromobility and small-scale autonomy.

Use of strong baselines: Comparing RT-DETR and RT-DETRv2 — both modern transformer-based detectors — provides an up-to-date reference point for zero-shot detection performance.

Dataset choice: The use of BDD100K, a large and diverse dataset with annotated time-of-day metadata, is appropriate and aligns well with the study’s goals.

Clear problem framing: The paper clearly articulates that lighting variation is a critical factor for pedestrian safety in ULEV applications, giving practical context to an otherwise academic evaluation.

Questions for the authors:

 - Which exact RT-DETR checkpoints were used?
 - Was any image preprocessing (spatial scaling, contrast normalization, gamma correction) applied to BDD100K frames before inference?
 - Were far-away pedestrians (few pixels tall) included in the recall computation, and how might they affect conclusions for slow-moving ULEVs?
 - Why was recall alone reported, without precision or F1 metrics?
 - Did you consider simple low-light normalization or histogram equalization baselines to test robustness?
 - Could results be stratified by pedestrian distance / pixel size to reveal performance dependencies?

Overall, the paper’s strengths lie mainly in its topic relevance and use of modern detectors, but the methodological contribution and experimental depth are limited.

**Weaknesses:**

Lack of novelty: The work performs a minimal benchmark using off-the-shelf models without architectural, training, or adaptation innovations.

Insufficient methodological detail: Missing descriptions of preprocessing, scaling, and checkpoint configuration preclude reproducibility.

Incomplete evaluation: Only recall is reported. No analysis of precision, false positives, or PR curves — leaving detection quality under-characterized.

Metric misuse and logical inconsistencies:
 - Reporting both recall and FNR as separate metrics is redundant (FNR=1−Recall).
 - The claim that validation recall is lower “as expected” is unfounded since no model training occurred.

Speculative interpretation: Statements about model “attentiveness” or operator fatigue are unsupported and detract from scientific clarity.

Neglect of temporal context: BDD100K includes video sequences, yet the study evaluates frames independently, ignoring potential improvements from temporal cues.

Weak significance: The main conclusion — that night scenes are harder — is already well-known; no actionable insight or methodological innovation is offered.

**Justification:**

While the paper addresses a relevant applied question, its scientific contribution is marginal. The work merely confirms a known issue—low-light degradation in pedestrian detection—using standard pretrained models and limited analysis. Missing methodological details and inconsistent reasoning undermine reproducibility and confidence in results.

Consequently, although the topic is of practical importance, the paper does not meet the standards of originality, soundness, or completeness expected for acceptance at a machine-learning conference. I therefore recommend Reject.

An LLM was used to help turn my notes into a readable review. The submission itself was not provided to the model, and the evaluations and conclusions are entirely my own.

---

### Meta-Review · Area_Chair_uYrc · 2025-10-29

**Recommendation:** Reject
**Confidence:** 4

**Metareview:**

This work has some strengths but also several significant weaknesses highlighted by all reviewers, including limited novelty, insufficient evaluation, and a lack of methodological details. As the authors did not use the rebuttal period to address these concerns, the reviewers leaned toward rejection, a conclusion with which I fully agree.

---

### Decision · Program_Chairs · 2025-11-05

**Decision:**

Reject

**Comment:**

Based on the reviewers and AC comments, the paper cannot be presented at the conference.